# A Novel Deep Dense Block-Based Model for Detecting Alzheimer's Disease

Selahattin Barış Çelebi [1,*] and Bülent Gürsel Emiroğlu [2]

1 Department of Computer Engineering, Graduate School of Natural and Applied Sciences, Kırıkkale University, 71450 Kırıkkale, Turkey

2 Department of Computer Engineering, Faculty of Engineering, Kırıkkale University, 71450 Kırıkkale, Turkey; bulentgursel@gmail.com

* Correspondence: sbariscelebi@gmail.com; Tel.: +90-543-427-0081

**Abstract:** Alzheimer's disease (AD), the most common form of dementia and neurological disorder, affects a significant number of elderly people worldwide. The main objective of this study was to develop an effective method for quickly diagnosing healthy individuals (CN) before they progress to mild cognitive impairment (MCI). Moreover, this study presents a unique approach to decomposing AD into stages using machine-learning architectures with the help of tensor-based morphometric image analysis. The proposed model, which uses a neural network built on the Xception architecture, was thoroughly assessed by comparing it with the most recent convolutional neural network (CNN) models described in the literature. The proposed method outperformed the other models in terms of performance, achieving an impressive average classification accuracy of 95.81% using the dataset. It also had very high sensitivity, specificity, accuracy, and F1 scores, with average scores of 95.41%, 97.92%, 95.01%, and 95.21%, respectively. In addition, it showed a superior classification ability compared to alternative methods, especially for MCI estimation, as evidenced by a mean area under the ROC curve (AUC) of 0.97. Our study demonstrated the effectiveness of deep-learning-based morphometric analysis using brain images for early AD diagnosis.

**Keywords:** Alzheimer's disease; transfer learning; image classification; deep learning; tensor-based morphometry

## 1. Introduction

A considerable proportion of elderly people worldwide are affected by AD, the most common form of dementia, which is a widespread neurological illness [1]. This illness causes a considerable decline in cognitive function, making it impossible for people to live conforming lives. As a result, sufferers require and depend on their loved ones' support to maintain their functioning [2]. AD is caused by a combination of genetic and environmental factors, including head injuries and chemical exposure. Memory loss, cognitive decline, communication problems, mood swings, and behavioral changes are just a few of the symptoms that define the illness. It is a disorder that progresses over time, has a pre-clinical stage, and its rates of progression can vary. The prognosis for people with AD is often poor, and the behavioral abnormalities brought on by the illness can make it difficult for patients and their caretakers to operate socially [3]. Modifications to the structural makeup of neurons are a part of AD's pathogenesis of AD. The microtubules in neurons act as conduits for the delivery of chemicals and nutrients to axons. TAU protein stabilizes these microtubules. In AD, the TAU protein undergoes chemical alterations, leading to its pairing with other TAU proteins. This process results in the formation of neurofibrillary tangles, causing neuronal collapse, cellular malfunction, and eventually cell death. Additionally, beta-amyloid plaques, known as senile plaques, and cerebrocortical atrophy are present, further hindering efficient information transmission [4,5]. The initial regions of the brain implicated in cognitive processes in AD are the hippocampus and medial temporal lobe.

These areas show a reduction in neuronal and synaptic density, contributing to cognitive decline. Magnetic resonance imaging (MRI) can visualize the atrophy of the hippocampus and other brain regions associated with memory processing and executive functions [6]. Figure 1 presents the prominent brain atrophy that is evident in individuals diagnosed with MCI and AD [7]. However, this phenomenon was not observed in healthy individuals.

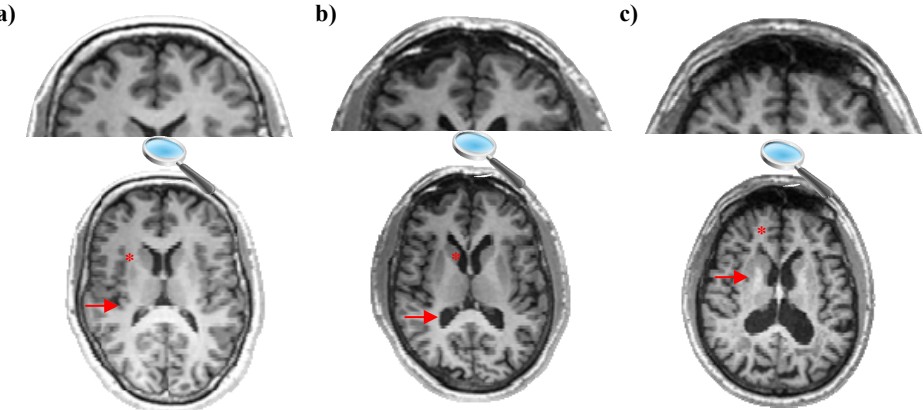

**Figure 1.** MRI scans of (**a**) CN, (**b**) MCI, and (**c**) AD adapted with permission from [7], Springer, 2018.

In Figure 1, the dimensions of the hippocampal tissue in MCI subjects, as highlighted with the red arrow, are smaller than those of cognitively normal (CN) subjects and further decrease in AD subjects. The magnitude of the ventricles undergoes a significant transformation, with an increase in size as the disease advances, as depicted with the red stars. Also, the decrease in the amount of gray matter in the cerebral cortex can be observed in magnified images of CN subjects compared to AD subjects.

MRI is a medical imaging modality that enables the production of high-resolution images that can visualize the differences between brain tissues [8]. Morphometric analysis is the process of obtaining quantitative data by evaluating and analyzing the geometric features of objects by processing MRI images. Through the amalgamation of MRI and morphometric analysis approaches, one can appraise the volume, morphology, and additional geometric characteristics of cerebral regions. In this way, its extensive utilization encompasses the diagnosis of cerebral disorders and the formulation of treatment strategies. Voxel-based morphometry (VBM) is the most widely known morphometric method. VBM calculates the density or concentration of gray matter in a voxel-wise manner [9]. Other morphometric methods, deformation-based morphometry (DBM) and tensor-based morphometry (TBM) use similar measurement techniques to characterize the differences in brain shape. DBM and TBM images are recorded in a common reference space and analyzed using the parameters of the deformation fields or measurements derived from them [10]. Surface-based morphometry (SBM) is another morphometric method that is used to analyze the surface properties of the cerebral cortex. The cerebral cortex can be modeled with a spherical model, and the features (thickness, fold depth, and surface area) in this model can be measured statistically [11]. Among the various methodologies available for neuroimaging analyses and clinical trials, TBM stands out as a highly reliable and objective measure with a significant capacity for high-throughput imaging [12].

Machine-learning (ML)-based systems have been successfully applied in various fields such as energy [13], robotics [14] health [15], and transportation [16]. These systems have demonstrated potential for assisting radiologists and physicians in the timely identification and categorization of AD via computer-aided diagnosis (CAD) systems [17]. Timely diagnosis and accurate analysis of brain atrophy are crucial, and the automated detection of brain atrophy can greatly contribute to these goals. Additionally, it can optimize radiologist efficiency by providing more accurate and efficient results. Various ML techniques, including feature extraction, deep networks, and transfer learning (TL), have been proposed

for the classification of AD and MCI [18–21]. Deep learning (DL), consisting of artificial neurons, has demonstrated superior performance in handling complex classification tasks compared to traditional ML methods [22,23]. CNNs, a specialized DL technique, have been widely utilized for AD diagnosis [24,25]. The primary objective of this present research is to devise a highly effective approach for the timely detection of patients in the MCI phase prior to their progression toward the AD stage. Encouraged by the aforementioned findings, we aimed to add to the detection of AD by combining DL and TBM methods.

This study introduced a novel method that utilizes cutting-edge architectures (Xception, VGG16, VGG19, and ResNet-50V2) to accurately detect AD and classify the different stages of dementia (MCI and AD). The proposed model, incorporating the Xception architecture-based deep dense block, was comprehensively evaluated via comparison with modern DL techniques.

The main aim of this article is to develop an effective method for early diagnosis of patients in the MCI stage before they progress to the AD stage. Existing studies on Alzheimer's diagnosis are limited, mostly focusing on traditional machine-learning-based methods for feature extraction from raw/semi-processed MRI images. In this study, we analyzed processed TBM images statistically. While morphometric images tend to provide better results for disease diagnosis compared to raw/semi-processed MRI images, they are challenging to interpret visually. Therefore, the use of DL-based methods, which enable automatic extraction of disease-specific features from difficult-to-interpret images, will address a significant research gap. The success of DL-based methods can assist physicians in diagnosing diseases using morphometric images. Hence, our goal was to contribute to AD detection by combining DL and TBM methods. Our study's results demonstrate that the proposed method, based on deep TL and TBM analysis, achieves accurate classification of three different classes and exhibits promising performance. The key highlights of our article include the following:

- Because MRI scans are inherently three-dimensional, they can be conceptualized as a stack of 2D MRI slices. From this stack, we selected the most informative slices for classification.
- MCI is a transitional stage between AD and CN. Therefore, it is difficult to diagnose. Therefore, in order to classify medical images, we employ the transfer-learning method using models trained on a large dataset.
- Transfer learning is used because there is limited data available, and it helps to reduce the costs of the learning process.
- Considering that morphometric images tend to yield more successful outcomes in disease diagnosis compared to raw or semi-processed MRI images, DL-based methods are employed for automatic feature extraction in the analysis of TBM images.

The rest of this paper is structured as follows: Section 2 provides a summary of related studies. Section 3 presents the CNN architecture and suggested work. Section 4 discusses the results and experiments, and Section 5 presents the results of this study.

## 2. Related Work

AD is a prevalent neurodegenerative disorder, and its early diagnosis is of the utmost importance. Consequently, various models and techniques have been introduced by the research community to facilitate the timely identification of AD. This section presents a review of the diverse deep-learning-based approaches utilized for the identification of AD.

Machine-learning methods, called traditional methods, were the first to appear [26,27]. A random forest ensemble classifier with adaptive hyperparameter tuning (HPT-RFE), a novel approach that performs faster than conventional ML algorithms, was employed by Kumari et al. They used MRI, FDG-PET, and PIB PET data from 102 participants to make a binary classification (NC/AD: 100%; NC: 91%; AD/MCI: 95%) of data from the ADNI database [28]. By transforming the 3D sMRI images from the Alzheimer's Disease Neuroimaging Initiative (ADNI) database into 2D matrices and then subjecting them to a series of processing steps, Gunawardena et al. were able to accurately detect AD at the

MCI stage with an SVM-based model and 96% using a deep-learning-based method. The CNN-based strategy outperformed the SVM-based method in detecting AD using sMRI data [29]. As an alternative to conventional information extraction techniques, Cruz et al.'s architecture, called HerstonNet, uses a 3D Resnet-based neural network regression model to extract significant characteristics from brain morphometric MRI data. By comparing HerstonNet to non-DL techniques, the consistency of the morphometric characteristics increased by 6.09% for volume, 21.73% for thickness, and 43.15% for mean curvature [30]. In a different study, Savas experimented with various CNN-based pre-trained architectures to categorize 2182 images taken from the ADNI database. The EfficientNetB3 design has the best accuracy, according to the results [31]. Turkson et al. used MRI scans to pre-train unsupervised convolutional spiking neural networks for binary classification. They subsequently created a pipeline architecture using the SNN output as a feed for a supervised deep CNN to perform AH/MCI/NC classification tasks. They compared the classification results produced using only a CNN with those produced using this architecture. The classification performance of the SNN + CNN pipeline design was significantly better than that of the CNN-based model alone [32].

For the multiclass classification of AD, Farooq et al. [33] presented deep-learning algorithms GoogLeNet, ResNet-18, and ResNet-152. Four classes (AD, LMCI, MCI, and CN) with MRI values of 33, 22, 449, and 45, respectively, were used in the experiments. ResNet-18 and ResNet-152 achieved accuracies of 98.88, 98.01, and 98.14%, respectively. Xia et al. [34], using AD (198), CN 229, and MCI (408) data, used 3D CLSTM to extract deep salient features and recognized 94.19% of AD cases. Using several CNN architectures, Ashraf et al. [35] realized fine-tuned features and reported a recognition rate of 99.05% for the diagnosis of AD. In [36], tissue segmentation was used to extract gray matter tissue from each patient. A binary class using the VGG architecture was then applied to produce a recognition rate of 98.73% for AD versus NC. Research that employs images processed for morphometric analyses is also available [37], in addition to research that uses raw or normalized MRI images as inputs to CNNs. A hybrid approach based on VBM and quantitative susceptibility mapping, an MRI technique for calculating magnetic susceptibility in tissues, was proposed by Sato et al. for the early identification of AD. Their methods outperformed the conventional VBM-based method in classifying the MCI and NC groups more effectively. However, the performance of their approach's MCI/AD classification (68%) was lower than anticipated [38].

## 3. Materials and Methods

This section presents materials and methods examined in this paper. Section 3.1 covers TBM. Section 3.2 discusses the data acquisition and pre-processing steps. Section 3.3 explains the deep-learning architectures used for feature extraction and the deep-learning-based classifier model used for classification. Section 3.4 focuses on TL. Finally, Section 3.5 elaborates on the variety of performance indicators employed to examine the efficacy of the suggested architecture.

### 3.1. Tensor-Based Morphometry

The TBM is an image analysis modality that evaluates structural disparities in the brain via gradients of deformation fields that are employed to align images. To evaluate the three-dimensional configurations of structural brain changes over time, discrete Jacobian maps were generated for each individual. Equation (1) shows the computation of the Jacobian matrix [39].

$$\mathbf{J} = \begin{bmatrix} \partial y_1 / \partial x_1 & \partial y_1 / \partial x_2 & \partial y_1 / \partial x_3 \\ \partial y_2 / \partial x_1 & \partial y_2 / \partial x_2 & \partial y_2 / \partial x_3 \\ \partial y_3 / \partial x_1 & \partial y_3 / \partial x_2 & \partial y_3 / \partial x_3 \end{bmatrix} \tag{1}$$

Selecting 2D slices from 3D images is a critical choice for image-processing applications. Therefore, 3D volumetric data were sampled at 5-pixel intervals in a range of improvements covering the hippocampus and temporal lobe. This is because tissues affected by

Alzheimer's are short sampled by manually selecting ranges to allow for the diversity of properties. This range of selection can be found with different selection techniques for different purposes [40].

Figure 2 shows the MRI images of individuals with (a) AD, (b) MCI, and (c) CN, respectively. Figure 2d–f are TBM images corresponding to images (a), (b), and (c), respectively. MRI images in the ADNI_1 database and their corresponding processed TBM images were determined by comparing subject IDs. TBM images are obtained by normalizing the MRI images in Figure 2 to a reference space. Unlike easily interpretable MRI images, TBM images tend to be more complex and challenging to visually interpret. This is because the TBM images capture the local morphological variations of individual subjects in relation to the group average. Notably, these morphometric images were primarily intended for statistical analysis at the group level rather than visual inspection.

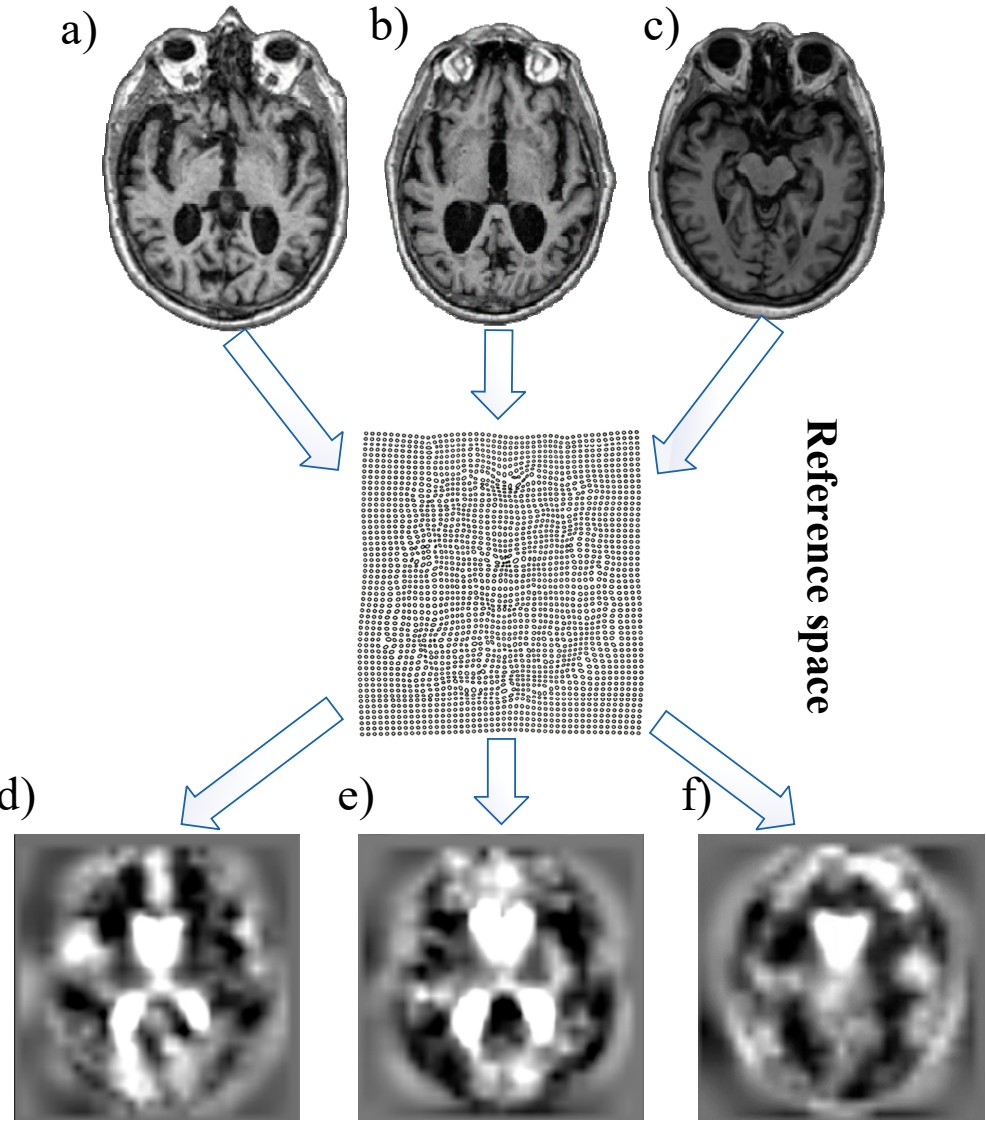

**Figure 2.** MRI scans: (**a**) axial MRI image of an AD; (**b**) axial MRI image of an MCI; (**c**) axial MRI image of a CN; (**d**) TBM image of figure (**a**); (**e**) TBM image of figure (**b**); (**f**) TBM image of figure (**c**).

In this study, TBM was used instead of raw or semi-processed MRI images. TBM has several of the following advantages over raw or semi-processed MRI images:

- It can precisely and quantitatively measure the shape and size properties of brain structures with TBM.
- It can compress and reduce the data volume in brain images. This can help with faster data processing.
- It can be used to compare structural changes between different individuals, groups, or time points.
- TBM has the ability to make statistical analyses of brain structures.

These advantages make tensor-based morphometry a powerful tool for deeper understanding of brain structures, diagnosis and follow-up of diseases, treatment planning, and neurological research [41].

### 3.2. Dataset and Pre-Processing

In this study, we gathered data from the ADNI, a worldwide research endeavor that actively promotes the study, analysis, and refinement of treatments for AD to arrest its progression. The ADNI datasets encompass datasets of various modalities, which can be beneficial to researchers in various ways for the early detection of AD. By providing standardized datasets, ADNI offers a means for researchers to conduct coherent research and disseminate compatible data to their counterparts worldwide. The TBM dataset used in this study consisted of 170 nifti files presented in nifti format and was obtained using the standard protocol presented on the ADNI Site. It consists of 28 AD subjects [mean age: $75.0 \pm 5.0$ years, 16 males (M)/12 females (F)], 88 patients with MCI [mean age: $73.8 \pm 5.4$ years, 47 (M)/41 (F)], and 54 CN subjects [mean age: $74.4 \pm 5.5$ years, 30 (M)/24 (F)]. Table 1 presents the demographic characteristics of the participants.

**Table 1.** Demographic characteristics of the dataset.

| Groups | CN | MCI | AD |
|---|---|---|---|
| Number | 54 | 88 | 28 |
| Gender (M/F) | 30/24 | 47/41 | 16/12 |
| Age (mean $\pm$ std) | $74.4 \pm 5.5$ year | $73.8 \pm 5.4$ years | $75.0 \pm 5.0$ year |

By converting volumetric 3D data into 2D representations, complex spatial information contained within the nifti format can be effectively extracted and leveraged using ML models. This conversion process facilitates the application of CNNs, which are used to learn hierarchical features from 2D image input. The 3D TBM images were uploaded to Google Colab. Subsequently, all the 3D TBM images were read using the Nibabel library in Python. First, the height and width of each nifti file are saved. Axial images were then sliced at 5-pixel intervals, starting from the 48th pixel and covering only the hippocampus and temporal lobe up to the 115th pixel. Consequently, 12 pieces of 2D axial brain slices were obtained for each subject. Considering that the hippocampus and temporal lobe are the regions most affected by AD, a pixel range of 48–115 was selected as the region of interest [42]. The following steps were applied to convert the TBM images from the 3D nifti format to the 2D png format (Figure 3).

As a result of the procedure shown in Figure 3, 2342 2D brain MR images of the subjects presented in Table 2 were obtained.

**Table 2.** Distribution of data utilized for training and testing the model.

| Data Set | AD | CN | MCI | Total Scans |
|---|---|---|---|---|
| Training set | 246 | 491 | 811 | 1548 |
| Validation set | 62 | 123 | 203 | 388 |
| Test set | 84 | 126 | 196 | 406 |

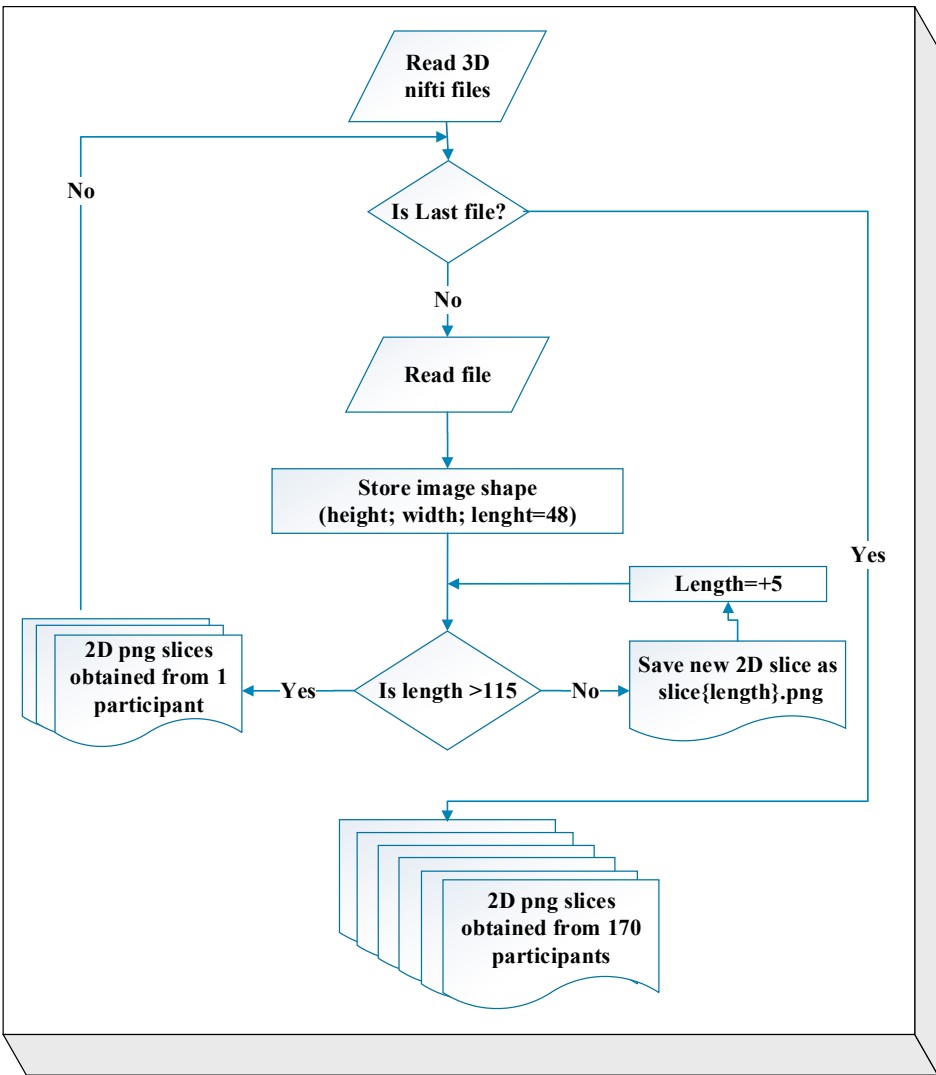

**Figure 3.** Dataset creation process.

*3.3. Proposed Classification Methods and Techniques*

3.3.1. CNN-Based Architectures

CNNs are deep neural networks that employ mathematical or linear operations known as "convolution". Multiple hidden layers, pooling layers, and output or fully connected (FC) layers form the base CNN architecture. These hidden layers are structured as a sequence of convolutional layers that contain filters, also known as kernels, which enable the network to perform image classification and predict patient diseases [43]. The pooling layer sequentially reduces the spatial size representation and hyperparameters. The pooling layer effectively minimizes the computational cost in CNNs and thus resolves the overfitting problem. The network also consists of ReLU activation functions. The mathematical formula for the ReLU is given in Equation (2) [44].

$$f_{\text{ReLU}} = max(0, x) \tag{2}$$

The CNN models are composed of a sequence of layers that employ filters to execute dimension reduction and feature extraction. CNNs are consequently referred to as "feature extractors", owing to their ability to efficiently extract significant features, analyze them independently, and classify them [45]. To generate a result, the FC layer was utilized in conjunction with the requisite collection of classes to execute a nonlinear modification of

the extracted features. This layer functions as an image classifier and analyzes and classifies the brain MRI images. The extraction of the feature vectors from the FC layers of the CNN was followed by their input into the units of the softmax layer for classification. The mathematical expression for the Softmax activation function can be represented as follows in Equation (3) [46]:

$$\sigma(\vec{z})_i = \frac{e^{z_i}}{\sum_{j=1}^{K} e^{z_j}} \tag{3}$$

where $\vec{z}$ is the input vector of the Softmax function, $Z_i$ is the input values of the input vector, $e^{z_i}$ the exponential function enforced to each element of the input vector, $K$ is the number of classes in the multiple classifiers, and $\sum_{j=1}^{K} e^{z_j}$ is the normalization operation.

### 3.3.2. Proposed Classifier

The features obtained from the base models of the CNN architectures used in this study were used as feature maps. The proposed classifier architecture (PPC) begins with a flattened layer that converts the feature maps into a vectorized format for the subsequent dense layer. The dense layer consists of 128 neurons with a ReLU activation function. To prevent overfitting, a dropout layer with a rate of 0.1 is employed. Subsequently, another dense layer with 64 neurons and the ReLU activation function was introduced, followed by another dropout layer at a rate of 0.1. A flattened layer is then used to flatten the data. The final layer, which is responsible for class predictions, consists of three neurons and employs a softmax activation function. The details of the neural network architecture model are listed in Table 3.

**Table 3.** Details of the proposed classifier architecture.

| Layer | Output Shape | Parameter |
|---|---|---|
| Flatten | (18,432) | 0 |
| Dense | (128) | 2,359,424 |
| dropout | (128) | 0 |
| dense | (64) | 8256 |
| dropout | (64) | 0 |
| flatten | (64) | 0 |
| dense | (3) | 195 |
| Total Parameter | | 23,229,355 |

### 3.4. CNN Training Based on Transfer Learning

Practical applications often require training a CNN using large datasets. However, in some cases, collecting large datasets comprising relevant problems can be challenging, and obtaining matching training and test data can be a complex process. Thus, the concept of TL has emerged. TL is one of the most effective ML methods and involves learning background knowledge to solve a problem and reusing it in other related problems. Initially, a base network is trained on a specific task using a relevant dataset and then transferred to a target task trained by a target dataset [47].

In this study, Xception [48], VGG-16, VGG-19 [49], and ResNet-50V2 [50] were utilized for feature extraction. These models were pre-trained on the ImageNet dataset. These models were pre-trained on the ImageNet dataset. The filters in the layers of the networks were employed to detect input features, such as colors, edges, lines, and local shapes. By utilizing the obtained outputs, it is possible to determine the class to which an input image belongs. In this study, pre-trained networks were used for the early diagnosis of AD. First, the FC layers were removed. The convolution and pooling layers outside this layer are responsible for extracting features. In addition, the feature-extracting layers of the pre-trained network were combined with these layers to classify a new class. Consequently, the training process can be established rapidly, requiring less training data than training a CNN. The extracted features were then used in a neural-network-based architecture to

perform the classification task (Figure 4). Figure 4 presents a flowchart of the proposed approaches for Alzheimer's diagnosis and classification based on the deep TL technique.

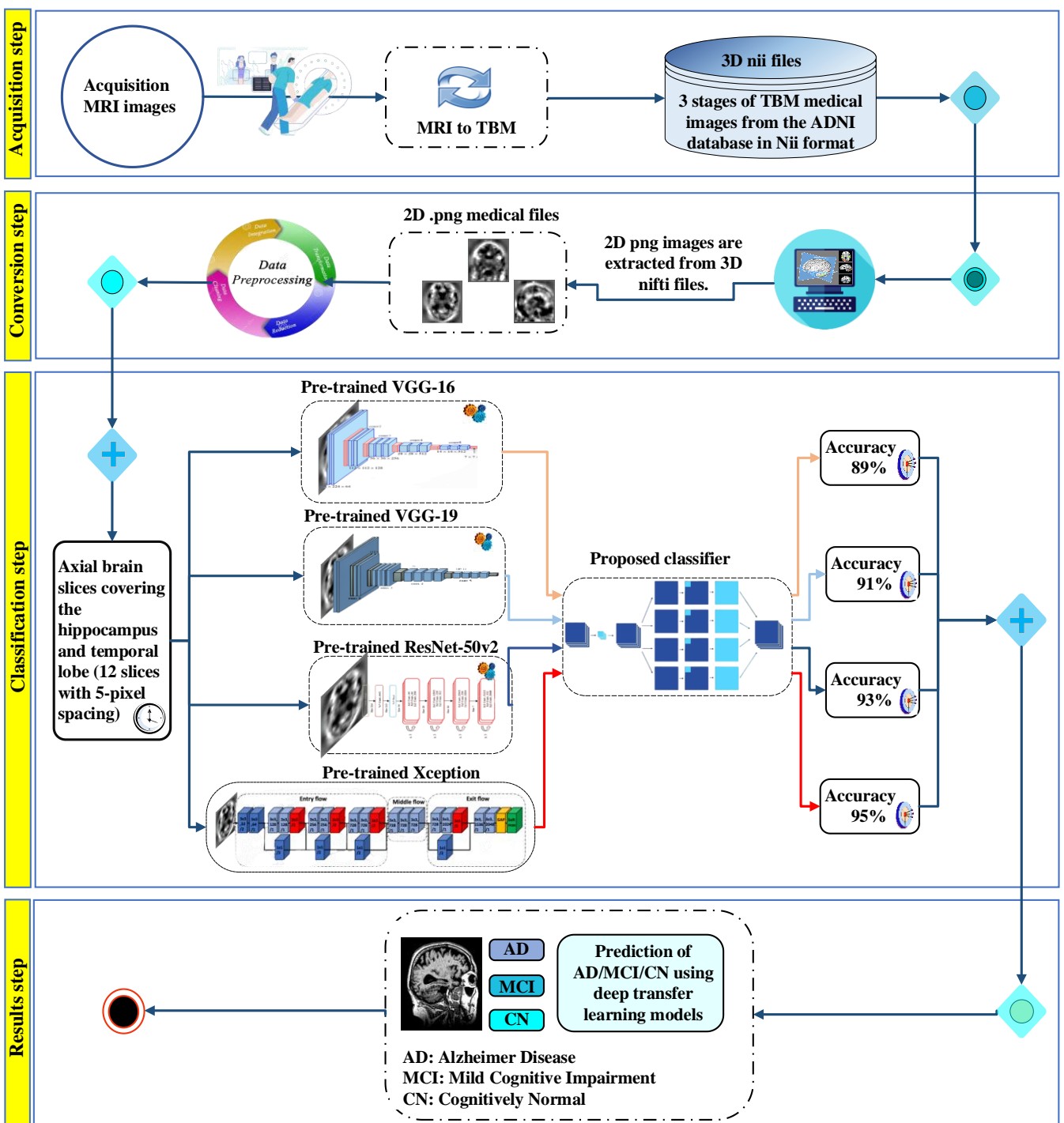

**Figure 4.** Flowchart of architectures utilizing pre-trained approach for early diagnosis of AD.

*3.5. Evaluation Metrics*

A confusion matrix is used to assess the efficacy of the classification problem. The matrix comprises four combinations of predicted and actual values: true positive (TP), true negative (TN), false positive (FP), and false negative (FN). Here, the accuracy (ACC-Equation (4)), precision (PRE-TPR-Equation (5)), sensitivity (recall- Equation (6)), specificity

(SPE-Equation (7)), F-Score (F-SCR-Equation (8)), and Negative Prediction Rate (NPR-Equation (9)) are defined as follows [51,52]:

$$ACC = \frac{TP + TN}{TP + TN + FP + FN} \tag{4}$$

$$PRE = \frac{TP}{TP + FP} \tag{5}$$

$$Recall = \frac{TP}{TP + FN} \tag{6}$$

$$SPE = \frac{TN}{TN + FP} \tag{7}$$

$$F - SCR = \frac{2 * PRE * Recall}{PRE + Recall} \tag{8}$$

$$NPR = \frac{TN}{TN + FN} \tag{9}$$

## 4. Results and Experiments

In this section, we outline the setup for the experiment and the results for the four models used in this study. Along with a competing model, we evaluated the performance of the suggested deep TL model.

### 4.1. Experimental Settings

The deep transfer-learning models tested in this manuscript are implemented using the Python 3.10.10 programming language and the Keras library within the TensorFlow 2.12 library. Keras is an open-source neural network library written in Python and has a high-end structure powered via both GPU and CPU. The hardware used for training the models is 64-bit, has a 4-core processor running at 2199 MHz, 32 GB of memory, and an Nvidia Tesla P100 GPU.

### 4.2. Hyperparameter and Optimization Techniques

The entire dataset is divided into approximately 66% training, 17% validation, and 17% testing. Validation and Test data are divided by taking close values in accordance with the literature [53]. The parameters that can impact model training are referred to as hyperparameters. To preserve the pre-learned filters, the convolutional base of the pre-trained architectures was entirely frozen, meaning that the weights of these layers remained unchanged during training. By changing the trainable parameters of the model to false, the convolutional base is frozen. The Adam optimizer was used to train our models for 200 epochs at a learning rate of $1 \times 10^{-3}$. The training hyperparameters are presented in Table 4.

**Table 4.** Hyperparameters used in all models.

| Training Parameters | | | | | | |
|---|---|---|---|---|---|---|
| **Learning Rate** | **Batch Size** | **Optimizer** | **Loss Function** | **Number of Epochs** | **Re-Scaling** | **Metrics** |
| $1 \times 10^{-3}$ | 32 | Adam | Categorical crossentopy | 200 | 1/0.255 | accuracy |

Throughout the training process, we applied various data augmentation techniques, including a zooming range of 20%, horizontal flipping, and a rotation of 45°. These

augmentation operations aimed to enhance the dataset and reduce the risk of overfitting (Table 5).

**Table 5.** Data augmentation parameters.

| Classes | Original Data | Augmentation Techniques | | |
|---|---|---|---|---|
| | | Rotation | Flipping | Zoom |
| | | 45° | Horizontal | (20%) |
| AD | 308 | 308 | 308 | 308 |
| CN | 614 | 614 | 614 | 614 |
| MCI | 1014 | 1014 | 1014 | 1014 |
| Total | 1936 | | 5808 | |

Subsequently, normalization was performed to facilitate learning. This technique aids in reducing computational intricacy by rescaling pixel values within the range of 0 to 1. For the multiclass classification problem, the "categorical_crossentropy" loss function and "accuracy" metrics are preferred. To prevent overfitting, the method implements an early stopping method that halts the training process if the accuracy of the validation dataset remains unchanged for a predetermined number of epochs.

### 4.3. Experimental Results

In this study, as mentioned in Section 3.3, TL was employed to train all DL models. We tested four deep TL models and evaluated their performances based on the indicators described in Section 3.4. The purpose of this research was to evaluate the effectiveness of the proposed deep TL model for the detection of AD and its early stages and to compare its performance with the most advanced CNN models reported in the existing literature. In order to achieve the main goals of this study, we carried out a comparative analysis of each model. To achieve this, we examined the test dataset's average F1 score, sensitivity, specificity, and accuracy scores for all models. According to the results presented in Table 6, the Xception + PPC model demonstrated the highest overall performance on the dataset, with an accuracy of 95.81%. Moreover, this model exhibited the highest sensitivity, specificity, precision, and F1 score, with values of 95.41%, 97.92%, 95.01%, and 95.21%, respectively. The Xception + PPC model also demonstrated excellent sensitivity, which is crucial for minimizing the misdiagnosis rate of volumetric changes in brain tissue. These findings indicate that Xception and PPC can effectively differentiate between Alzheimer's stages. The Xception architecture uses a technique called separable convolution. This technique is a more efficient computational method than traditional convolution. Depthwise separable convolutions offer superior expressiveness and productivity compared with classical convolutions. By incorporating depth-wise separable convolutions, the Xception model becomes highly proficient at learning distinct and high-level features that may be overlooked by simpler models. Another successful model is the ResNet-50V2 + PPC model, which achieves an accuracy of 93.35%. Additionally, the sensitivity, specificity, and F1 scores reached 92.81%, 96.59%, and 92.57%, respectively.

**Table 6.** The performance evaluation and comparison of deep TL architectures using various indicators for the purpose of Alzheimer's diagnosis.

| Models | Training Accuracy | Validation Accuracy | Test (Macro Avg) | | | | |
|---|---|---|---|---|---|---|---|
| | | | Accuracy | Specificity | Precision | Sensitivity | F1-Score |
| VGG-16 + PPC | 94.98% | 90.55% | 89.66% | 94.94 | 87.94 | 89.04 | 88.14 |
| VGG-19 + PPC | 95.63% | 93.39% | 91.63% | 95.89 | 90.07 | 90.89 | 90.32 |
| ResNet-50V2 + PPC | 97.54% | 94.01% | 93.35% | 96.59 | 92.41 | 92.82 | 92.57 |
| Xception + PPC | 97.60% | 95.89% | 95.81% | 97.92 | 95.01 | 95.41 | 95.21 |

Figure 5 shows the accuracy achieved in the test datasets for all models. The test accuracy depicted in Figure 5 is computed by dividing the number of correctly classified patients (CN, MCI, and AD) by the total number of patients. Figure 5 clearly shows that the Xception + PPC model outperformed the other three models.

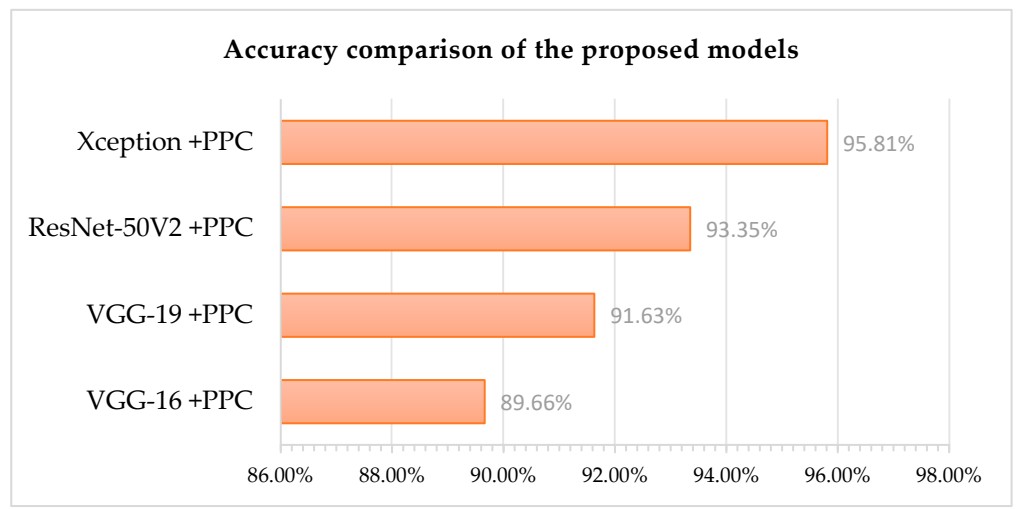

**Figure 5.** Accuracy ratios in the test set for all the models.

Table 7 presents the performance of the models by class. The best results are bolded. For training purposes, 1936 medical images were utilized, while 406 additional images were designated for testing. The analysis included three classes: AD, MCI, and CN. From the results table, it is evident that the Xception + PPC model exhibits strong performance across all classes. The model achieved an average precision, sensitivity, and F1 score of 0.95. The model attained a precision of 0.98 and demonstrated a good sensitivity of 0.97 for the MCI classes. These findings demonstrate the model's capability to attain a high level of accuracy in diagnosing the disease, particularly in the early stages of Alzheimer's disease, such as the MCI stage. For the macro-average scores of all evaluation metrics, it is evident that the Xception + PPC model outperformed the other models.

**Table 7.** Performance of different deep TL architectures with various metrics.

| Models | Class | Precision | Sensitivity | F1-Score | Accuracy |
|---|---|---|---|---|---|
| VGG-16 + PPC | AD | 0.78 | 0.93 | 0.85 | |
| | CN | 0.91 | 0.79 | 0.85 | |
| | MCI | 0.95 | 0.95 | 0.95 | 89.66% |
| | Macro Average | 0.88 | 0.89 | 0.88 | |
| VGG-19 + PPC | AD | 0.80 | 0.90 | 0.85 | |
| | CN | 0.95 | 0.87 | 0.91 | |
| | MCI | 0.95 | 0.95 | 0.95 | 91.63% |
| | Macro Average | 0.90 | 0.91 | 0.90 | |
| ResNet-50V2 + PPC | AD | 0.88 | 0.93 | 0.90 | |
| | CN | 0.94 | 0.90 | 0.92 | |
| | MCI | 0.95 | 0.96 | 0.96 | 93.35% |
| | Macro Average | 0.92 | 0.93 | 0.93 | |
| **Xception + PPC** | **AD** | **0.92** | **0.94** | **0.93** | |
| | **CN** | **0.95** | **0.95** | **0.95** | |
| | **MCI** | **0.98** | **0.97** | **0.97** | **95.81%** |
| | **Macro Average** | **0.95** | **0.95** | **0.95** | |

Figure 6 shows the confusion matrix, which provides a detailed overview of the class-wise results for the Xception + PPC model. By examining the confusion matrix, we assessed the number of correctly classified and misclassified images for the specific classes. From the confusion matrix, we can infer that the Xception + PPC model misclassified only seventeen images from the test dataset. Of the 406 tests, 389 were accurately classified by the model, demonstrating a high accuracy. Therefore, based on the evaluation metrics accomplished by the proposed model, it can be inferred that Xception + PPC outperformed the other models in all aspects.

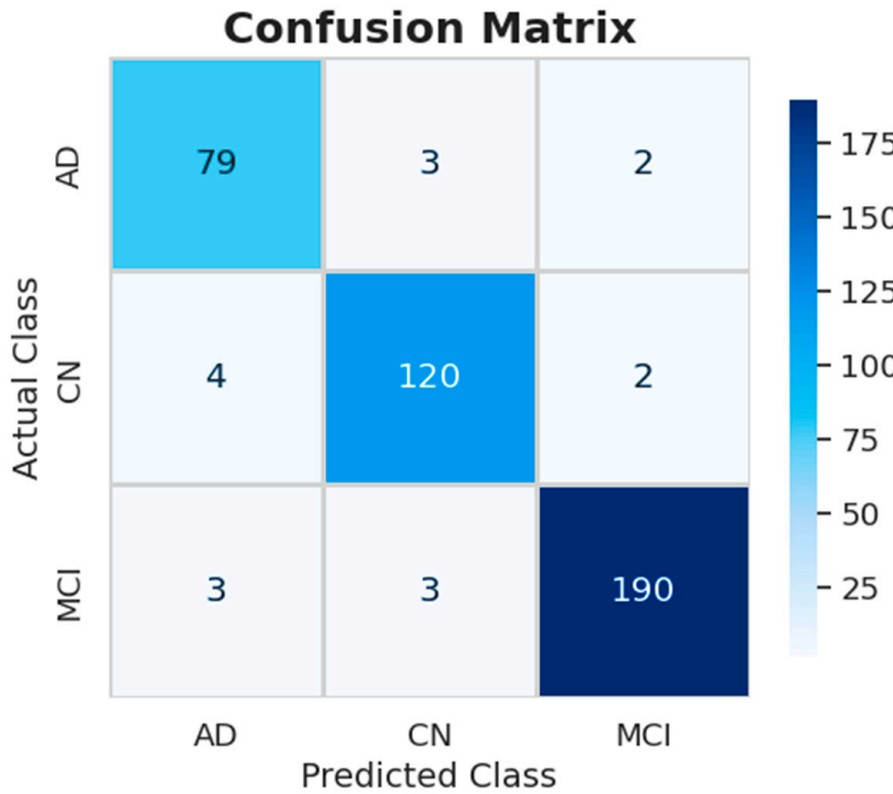

**Figure 6.** Confusion matrix from Xception + PPC model.

We want the area to be 1 in an ideal receiver operating characteristic (ROC) curve and aim to move away from the FPR value as the TPR value increases. Within this context, the suggested approach demonstrated superior classification capability compared to alternative methods, as evidenced by an average AUC of 0.97, specifically achieving an 0.98 AUC for MCI prediction (Figure 7). The effectiveness of this method becomes apparent when the confusion matrix is examined.

### 4.4. Performance Evaluation in Relation to Baseline Models

Our study aimed to evaluate the effectiveness of the proposed deep TL model for diagnosing AD and its early stages and compare its performance with that of state-of-the-art CNN models reported in the literature. The classification performance of the models was assessed in terms of accuracy, sensitivity, precision, and the F1 score. The proposed Xception + PPC model exhibited superior accuracy compared with the baseline model, achieving a 6.65% increase. Furthermore, the VGG-16 + PPC, VGG-19 + PPC, and ResNet-50V2 + PPC models demonstrated accuracy improvements of 3.7, 4.19, and 5.42%, respectively. The incorporation of multiple dense layers in the PPC contributed to enhanced learning ability and accuracy. PPC also exhibited improved detection rates and stability for AD classification. The baseline models showed lower performance on the test set, with accuracies ranging from 85% to 89% as well as lower sensitivity and

F1 scores. Several factors, including dataset variations, overfitting, and challenges in feature extraction from TBM images, contribute to the underperformance of base models. A comparison of the computational costs revealed that the Xception + PPC model had a longer training time (3130.86 s) than the base model. However, our primary focus was on improving the accuracy of the method, which showed significant enhancement after the incorporation of PPC. Overall, our proposed method outperformed the baseline models in terms of classification performance and demonstrated potential for accurate AD diagnosis (Table 8).

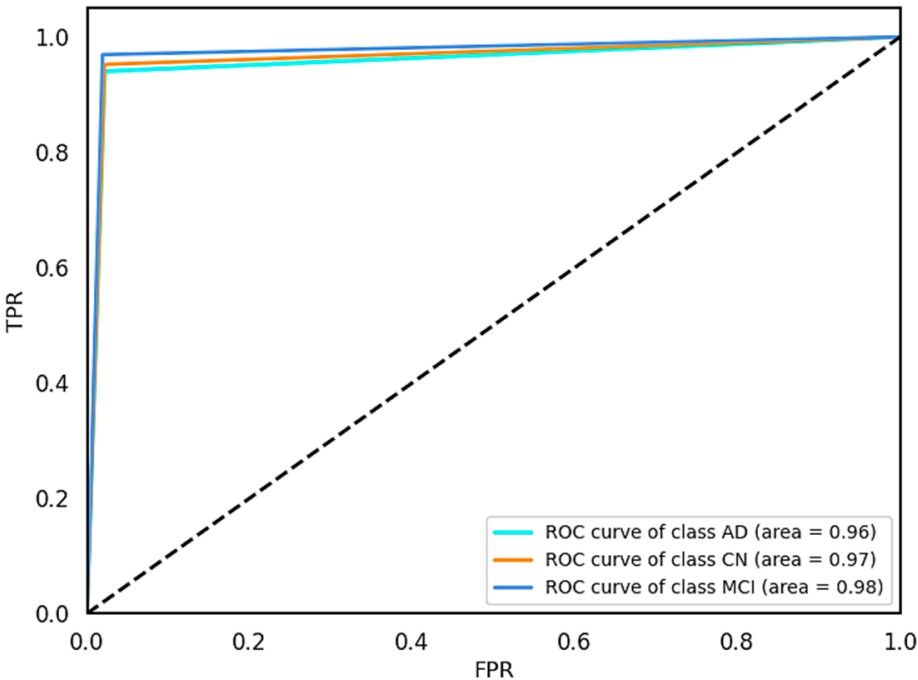

**Figure 7.** ROC curve from Xception + PPC model.

**Table 8.** Performance evaluation of baseline models and proposed approaches.

| Models | Accuracy | Precision | Sensitivity | F1-Score | Training Time (s) | Average Time Per Epoch (s) | Test Time (s) |
|---|---|---|---|---|---|---|---|
| VGG-16 (Baseline) | 85.96% | 84.08 | 85.49 | 84.55 | 2594.75 | 12,97 | 1.39 |
| VGG-19 (Baseline) | 87.44% | 86.02 | 86.98 | 86.45 | 3174.44 | 15.87 | 4.77 |
| ResNet-50V2 (Baseline) | 87.93% | 86.60 | 87.13 | 86.81 | 2749.96 | 13.74 | 5.52 |
| Xception (Baseline) | 89.16% | 88.10 | 88.76 | 88.35 | 2982.52 | 14,91 | 4.72 |
| VGG-16 + PPC | 89.66% | 87.94 | 89.04 | 88.14 | 3192.16 | 15.96 | 1.91 |
| VGG-19 + PPC | 91.63% | 90.07 | 90.89 | 90.32 | 2844.81 | 14.22 | 4.88 |
| ResNet-50V2 + PPC | 93.35% | 92.41 | 92.82 | 92.57 | 2668.44 | 13.34 | 3.38 |
| Xception + PPC | 95.81% | 95.01 | 95.41 | 95.21 | 3130.86 | 15.65 | 5.04 |

### 4.5. Comparison with Related Works

Morphometric-based studies have been limited to the literature. Research related to the application of deep-learning methods for the early detection of AD is scarce. Furthermore, existing studies have predominantly focused on VBM analyses, with only a few studies comparing TBM images. The effectiveness of the suggested model based on the Xception + PPC architecture was compared with that of other competing models. Many studies have utilized the same dataset for classification purposes; therefore, we selected the ADNI database for our research. Accuracy was used as the major parameter to evaluate the

outcomes of classification. Table 9 presents a comparison of the suggested model using the same dataset as other comparative studies in the literature, demonstrating its superior performance with a 95.81% accuracy rate in the three-class dataset. In addition, most studies have focused on binary classifications. To the best of our knowledge, no study has focused on the early diagnosis of AD using deep TL-based TBM analysis for multiclass classification. The proposed model exhibits the capability to effectively address a three-class problem.

**Table 9.** Comparing the suggested model with cutting-edge models on the same dataset.

| Reference | Biomarker | Database | Method(s) | ACC (AD, MCI, CN) | Participant | Approach |
|---|---|---|---|---|---|---|
| [32] | MRI | ADNI | SNN + CNN | NC/AD: 90.15%<br>MCI/AD: 87.30%<br>NC/MCI: 83.90%<br>NC/MCI: 87%<br>MCI/AD: 68% | 450 | ROI |
| [54] | VBM | ADNI | SegNet + ResNet-101 | AD/CN/MCI: 96% | 240 | ROI |
| [55] | VBM | ADNI | LeNet<br>AlexNet<br>VGGNet<br>GoogLeNet<br>ResNet | NC/AD: 93.83%<br>NC/AD: 96.22%<br>NC/AD: 96.08%<br>NC/AD: 97.15%<br>NC/AD: 94.60% | 479 | 3D subject level |
| [56] | VBM | ADNI | CNN | MCI: 80.9% | 188 | 3D subject level |
| In this study | TBM | ADNI | Xception + PPC | AD/CN/MCI: %95.81 | 170 | ROI |

Despite utilizing a unified architecture, they achieved the highest accuracy rate of 87% using raw MRI data [32]. In this study, a higher success rate was achieved compared to methods that utilized raw datasets in parallel with other related studies. Similarly, it has been observed that morphometric methods achieved higher accuracy rates than raw MRI datasets [54,55]. In [55], the VGGNet base architecture was used for VBM-based analysis, resulting in 96% accuracy. In this study, the accuracy rates of 85% and 87% were obtained using the baseline VGGNet16 and VGGNet19, respectively. This finding suggests that a VBM-based analysis using base models may yield more successful results than a TBM-based analysis.

*4.6. Strengths and Limitations*

So far, studies have generally focused on binary classification problems for Alzheimer's diagnosis. This study specifically addresses the multiple classification problem of AD/MCI/CN. This study was conducted using a publicly available dataset. To mitigate overfitting issues and extract disease-specific features, data augmentation was employed due to the limited number of images in the dataset.

The proposed model utilizes a deep neural network and does not require separate feature extraction. Various databases are utilized in the literature for AD diagnosis. However, collecting data from entire neuroimaging databases can be challenging. Additionally, neuroimages are processed using different methods. In this study, an ROI-based method was implemented to improve the accuracy rate. Furthermore, performance enhancements were attempted via transfer learning.

Morphometric images may possess higher dimensions compared to normal MRI images, thereby requiring increased memory and processing power. Furthermore, manually labeling such datasets can be arduous. Hence, expert knowledge and time-consuming manual procedures may be necessary for generating labeling data.

Despite CNN's favorable performance in medical image analysis, there are still lingering issues. Limited data availability is particularly problematic in the field of medical image processing. To overcome this, a large database was preferred for this study.

Although the transfer-learning method employed in this study boasts numerous advantages, a neural network based on the complex structure of the Xception architecture presents challenges. These complex model structures can affect the method's applicability, including training, hyperparameter tuning, and computational resource requirements. To overcome these limitations, successful adaptive methods such as the Adam optimizer were utilized.

## 5. Conclusions

The primary objective of this study is to develop an automated DL method for the early detection of AD. Determining the disease stage presents a significant challenge because of the high similarity between AD stages. To overcome this challenge, we employed morphometric methods and conducted experiments involving three types of classification. All images were preprocessed using image-processing techniques. For AD detection using TBM images, we adopted four popular deep-learning architectures based on the deep TL technique. The last layer of examined architectures was completed with deep dense blocks and softmax layers to enhance classification performance. Specifically, our proposed model, based on the Xception architecture, utilizes depth-wise separable convolution, enabling the efficient learning of noticeable and high-level features. The incorporation of a deep dense block further enhances the performance of the model. Normalizations of data, data augmentation, and dropouts were employed to mitigate overfitting, whereas the Adam optimizer ensured fast learning. The proposed model obtains an impressive overall classification accuracy of 95.81% for the dataset used, clearly outperforming other models in terms of performance. Our model exhibits superior classification accuracy compared to existing models. In future work, we intend to expand the dataset by incorporating additional TBM brain image data, including sagittal and coronal images, while maintaining performance standards. Moreover, we plan to enhance our architecture further by conducting experiments using different parameter settings.

**Author Contributions:** Conceptualization, S.B.Ç. and B.G.E.; methodology, S.B.Ç. and B.G.E.; software, S.B.Ç.; validation, B.G.E.; formal analysis, B.G.E.; investigation, B.G.E.; resources, S.B.Ç.; writing—original draft preparation, S.B.Ç. and B.G.E.; writing—review and editing, S.B.Ç.; visualization, S.B.Ç. and B.G.E.; supervision, B.G.E.; project administration, B.G.E.; funding acquisition, B.G.E.; All authors have read and agreed to the published version of the manuscript.

**Funding:** This research received no external funding.

**Institutional Review Board Statement:** Ethical review and approval were waived for this study, as this study only makes use of publicly available data.

**Informed Consent Statement:** Not applicable.

**Data Availability Statement:** The data that support the findings of this study were obtained from the ADNI (publicly available at http://adni.loni.usc.edu/, accessed on 6 July 2022).

**Acknowledgments:** Data collection and sharing for this project was funded by the Alzheimer's Disease Neuroimaging Initiative (ADNI) (National Institutes of Health Grant U01 AG024904) and DOD ADNI (Department of Defense award number W81XWH-12-2-0012). ADNI is funded by the National Institute on Aging, the National Institute of Biomedical Imaging and Bioengineering, and through generous contributions from the following: AbbVie, Alzheimer's Association; Alzheimer's Drug Discovery Foundation; Araclon Biotech; BioClinica, Inc.; Biogen; Bristol-Myers Squibb Company; CereSpir, Inc.; Cogstate; Eisai Inc.; Elan Pharmaceuticals, Inc.; Eli Lilly and Company; EuroImmun; F. Hoffmann-La Roche Ltd. and its affiliated company Genentech, Inc.; Fujirebio; GE Healthcare; IXICO Ltd.; Janssen Alzheimer Immunotherapy Research & Development, LLC.; Johnson & Johnson Pharmaceutical Research & Development LLC.; Lumosity; Lundbeck; Merck & Co., Inc.; Meso Scale Diagnostics, LLC.; NeuroRx Research; Neurotrack Technologies; Novartis Pharmaceuticals Corporation; Pfizer Inc.; Piramal Imaging; Servier; Takeda Pharmaceutical Company; and Transition Therapeutics. The Canadian Institutes of Health Research provides funds to support ADNI clinical sites in Canada. Private sector contributions are facilitated by the Foundation for the National Institutes of Health (www.fnih.org, accessed on 6 July 2022). The grantee organization is the Northern

California Institute for Research and Education, and this study is coordinated by the Alzheimer's Therapeutic Research Institute at the University of Southern California. ADNI data are disseminated by the Laboratory for Neuro Imaging at the University of Southern California.

**Conflicts of Interest:** The authors declare no conflict of interest.

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
