# Peer review of "A Novel Deep Dense Block-Based Model for Detecting Alzheimer’s Disease"

_applsci, doi:10.3390/app13158686_

Round 1

Reviewer 1 Report

The presented manuscript proposed a deep-learning method to Detect Alzheimer's Disease from MRI images. They present a morphological method on the images and then classify the images into 3 classes using transfer learning and achieved an average classification accuracy of 95.81% using the ADNI dataset. The main shortcoming of the manuscript is that the work does not state any research questions, problem statements, or contributions. The manuscript is well written, but there are some vague points in the manuscript that have to be addressed and the manuscript is not appropriate for publication. The following comments and questions would require clarification before acceptance.

 *** Detailed comments and questions ***

1-    The work does not provide a clearly defined aim nor research questions that it aims to answer. The question which the authors investigated in this study has to be clarified.

2-    The authors have used the TBM method among the morphologic methods. What is the reason for using this method? What is the size of the tensors considered in using this method?

3-    In Figure 2, it is suggested to add images before applying TBM so that the changes can be recognized. Please modify this.

4-    The authors mentioned “12 pieces of 2D axial brain 197 slices were obtained for each subject”, What were the criteria for choosing 12 slices?

5-    A lot of additional facts and explanations have been given about the networks, especially the CNN network, that it is not necessary to deal with this issue in detail in this manuscript.

6-    The structure introduced for the classifier in Table 3 is not the structure of a CNN. What did the authors mean by CNN? It is a neural network, not a CNN. Please modify this.

7-    In the proposed evaluation Metrics, what is the difference between PRE and PPR, and why the authors are using both these metrics? What additional information can each of them give us?

8-    The authors mentioned “The dataset is divided into training (80%), validation (20%), and test (~21%) sets” which is not possible and the total dataset is not 121%. What did the authors mean by this? Please justify it.

9-    Due to the transfer learning technique used in the manuscript, there is no need for data augmentation. What is the reason for doing data augmentation that the authors mentioned in Table 5?

Reviewer 2 Report

Reviewers understand that Çelebi et. al have presented a manuscript entitled "A Novel Deep Dense Block-Based Model for Detecting Alzheimer's Disease". Reviewers would like to request authors to kindly make following changes in their manuscript.

1) Kindly improve quality of all the figures and provide high resolution figures in your manuscript. Provide naming to different parts of figure 1 and 2. Figure 3 and 4 details are not readable and kindly increase the font size of both the figures sub-part description. Also pictorial details of figure 4 are non-understandable, please provide another figure with larger sub-details. 

2) Kindly provide details about limitations of your currently used methods. Also provide steps you took to work on these limitations to improve them in your current studies.

Reviewer 3 Report

This paper is very well written and contributes develop an automated deep learning method for the field of the early detection of Alzheimer's disease, which can promote the development of health care. And the proposed scheme outperforms the state of the arts and can be applied to the diagnostic field of neurological diseases and has a broad prospect. However, there are some problems, which must be solved before it is considered for publication.

1. When explaining each variable in formula (3), it should be consistent with the formula, otherwise the readability of the expression will be affected. For example, Zi is more appropriately changed to Zi.

2. The completeness of sentences in the manuscript should be carefully checked by the author. For example, on line 241, there is a lack of a period at the end of the sentence.

3. The structure of Section 4 is not reasonable, and the author's arrangement of Section 4.1 is too brief, so the author is required to rearrange the structure of the article or describe it in detail.

4. The final model proposed in the manuscript is the Xception+PPC structure. Why did the author introduce too much VGG-16, VGG-19, and ResNet-50V2 architectures in Section 3.3, and combine the flow charts of these methods in Figure 4 to describe it? The innovation and contribution of this article should be highlighted when introducing the principle.

5. In 3.3.4, the Xception architecture adopted a deep tree structure and separable convolution. However, in the analysis of the experimental results, line 357, the increase in computational efficiency is due to the use of general convolution instead of deep separable convolution in the Xception model. Whether the logic before and after is contradictory requires the author to give a reasonable explanation.

Minor editing of English language required.

Round 2

Reviewer 1 Report

The authors have answered all of my questions and the paper has been improved. Therefore, it can be accepted for publication.

Reviewer 3 Report

Thank the authors for their efforts. The authors have adequately addressed all my concerns in the review, and did a good job to revise and improve the paper. The paper now is suitable for publication in its current form.